# Mutation of Tyrosine Sites in the Human Alpha-Synuclein Gene Induces Neurotoxicity in Transgenic Mice with Soluble Alpha-Synuclein Oligomer Formation

**DOI:** 10.3390/cells11223673

**Published:** 2022-11-18

**Authors:** Louise Berkhoudt Lassen, Maj Schneider Thomsen, Elisa Basso, Ernst-Martin Füchtbauer, Annette Füchtbauer, Tiago Fleming Outeiro, Poul Henning Jensen, Torben Moos

**Affiliations:** 1DANDRITE, Department of Biomedicine, Aarhus University, 8000 Aarhus, Denmark; 2Neurobiology Research and Drug Delivery, Department of Health Science and Technology, Aalborg University, 9220 Aalborg, Denmark; 3Center for Biostructural Imaging of Neurodegeneration, Department of Experimental Neurodegeneration, University Medical Center Göttingen, 37075 Göttingen, Germany; 4Department of Molecular Biology and Genetics, Aarhus University, 8000 Aarhus, Denmark; 5Max Planck Institute for Multidisciplinary Sciences, 37075 Göttingen, Germany; 6Translational and Clinical Research Institute, Faculty of Medical Sciences, Newcastle University, Framlington Place, Newcastle Upon Tyne NE2 4HH, UK

**Keywords:** Parkinson’s disease, alpha-synuclein, neurodegeneration, tyrosine, serine, transgene, behavior, aggregates, cortico-spinal tract, oligodendrocyte

## Abstract

Overexpression of α-synuclein with tyrosine mutated to phenylalanine at position 125 leads to a severe phenotype with motor impairment and neuropathology in *Drosophila*. Here, we hypothesized that tyrosine mutations would similarly lead to impaired motor performance with neuropathology in a rodent model. In transgenic mice (ASO), tyrosines at positions 125, 133, and 136 in human α-synuclein were mutated to phenylalanine and cloned into a Thy1.2 expression vector, which was used to create transgenic mouse lines on a mixed genetic background *TgN(Thy-1-SNCA-YF)4Emfu* (YF). The YF mice had a decreased lifespan and displayed a dramatic motor phenotype with paralysis of both hind- and forelegs. Post-translational modification of α-synuclein due to phosphorylation of serine 129 is often seen in inclusions in the brains of patients with α-synucleinopathies. We observed a slight but significant increase in phosphorylation of serine 129 in the cytosol in YF mice compared to age-matched human α-synuclein transgenic mice (ASO). Conversely, significantly decreased phosphorylation of serine 129 was seen in synaptosomes of YF mice that also contained higher amounts of soluble oligomers. YF mice deposited full-length α-synuclein aggregates in neurons widespread in the CNS with the main occurrence in the forebrain structures of the cerebral cortex, the basal ganglia, and limbic structures. Full-length α-synuclein labeling was also prominent in many nuclear regions of the brain stem, deep cerebellar nuclei, and cerebellar cortex. The study shows that the substitution of tyrosines to phenylalanine in α-synuclein at positions 125, 133, and 136 leads to severe toxicity in vivo. An insignificant change upon tyrosine substitution suggests that the phosphorylation of serine 129 is not the cause of the toxicity.

## 1. Introduction

α-synuclein is a 140-amino acid protein prone to aggregate, which leads to neurodegeneration with synaptic and mitochondrial dysfunctions, endoplasmic reticulum stress, and adjoining neuroinflammation [1]. Accompanying, these events is impairment in the degradation of protein aggregates via the proteasomal or autophagic pathways, thereby contributing to the pathogenesis of neurodegenerative disorders, such as Parkinson’s disease and other α-synucleinopathies [1].

Post-translational modifications of α-synuclein have been found to occur with phosphorylation of serine 129 attracting the most interest, as it is frequently found in aggregated α-synuclein-containing deposits in patients with α-synucleinopathies [2,3,4]. Serine phosphorylation is important for the functioning of α-synuclein but also for its potential toxicity; several studies underline the importance of serine phosphorylation of α-synuclein for its participation in events occurring within the cell, e.g., aggregation, nuclear translocalization, and degradation [5,6,7].

Several kinases are responsible for serine 129 phosphorylation of α-synuclein [3]. Efficient modification of serine 129-α-synuclein by the kinase casein kinase (CK) 1 requires phosphorylation of tyrosine 125 as a priming event [8]. Upon tyrosine phosphorylation in vitro, phosphorylation of serine 129 increases significantly, which is mediated by CK 1 [9], known to constitute the only target with a preference for tyrosine 125-α-synuclein over serine 129-α-synuclein [10,11]. In yeast, three tyrosines of α-synuclein, at positions 125, 133, and 136, situated near serine 129, represent potential chemical and physical hotspots for interaction with other proteins and internal regulation of each post-translational modification [9].

*Drosophila* flies overexpressing α-synuclein with 125 tyrosine mutated to phenylalanine show impaired climbing ability and increased neuropathology suggestive of neuroprotective action of canonical α-synuclein with phosphorylation of tyrosine at position 125 [12]. The tyrosine phosphorylation decreases with age in both *Drosophila* and man. Tyrosine phosphorylation of α-synuclein is decreased in patients with Lewy-body dementia compared to non-diseased controls [12,13].

Correspondingly to observations made in *Drosophila*, we hypothesized that α-synuclein with tyrosines mutated to phenylalanine would lead to impaired motor performance and neuropathology in a mammalian model. Serine 129 phosphorylation of α-synuclein is strictly dependent on the presence of phosphorylated tyrosine 133 and tyrosine nitration and phosphorylation could have opposing roles [9,14]. To avoid the possibility of both phosphorylation and nitration of α-synuclein, we substituted the three C-terminal tyrosines with phenylalanine using both cellular and mouse models. In the present study, we demonstrate that the substitution of three tyrosines to phenylalanine at positions 125, 133, and 136 leads to high dose-dependent toxicity in vivo, expressed as impaired motor skills and neuropathology. We also show that the phosphorylation status of serine 129 decreased in synaptosomes upon tyrosine substitution, suggesting that the serine 129 phosphorylation is not an apparent cause of toxicity.

## 2. Materials and Methods

### 2.1. Generation of Transgenic Mouse Lines

Wild-type human α-synuclein with tyrosine at positions 125, 133, and 136 (ASO) was mutated to phenylalanine and cloned into the Thy1.2 expression vector generously donated by Dr. Masliah [15]. The transgene, including the promoter region of the Thy1 gene, was used to create three transgenic mouse lines on a mixed B6D2F1 genetic background: *TgN(Thy-1-SNCA-YF)4Emfu* (herein named YF), *TgN(Thy-1-SNCA-YF)2Emfu* (herein named: YF^low1^), and *TgN(Thy-1-SNCA-YF)1Emfu* (herein named: YF^low2^). The coding sequence of human α-synuclein, in which the three tyrosine residues at positions 125, 133, and 136 were substituted by phenylalanine, was inserted into the XhoI cloning site, which is situated after exon 1a, 1b and the first part of exon 2 of ThyI, and before the last part of exon 4 of the Thy1 gene, hence, ensuring that YF α-synuclein is under the control of the Thy1 promoter. The vectors were linearized and injected into mice pronuclei inserted into a foster mother giving rise to mouse pups, which, upon positive results from PCR of insertion of the vector, were considered the founders of the individual YF^low^ mouse lines. In the YF line, only founder mice were used for experiments, because the offspring were too weak to breed.

All experiments on mice were performed according to the Danish Animal Welfare Agency requirements (approval number 2017-15-0201-01203) and in accordance with national and international guidelines for the care and use of animals. Mice had constant water and food supply and were held under temperature-constant conditions with a 12-h light/dark cycle. Due to a physically weak phenotype, the YF mice were given peanut butter to help to maintain body weight. WT littermates to YF mice were kept together with YF mice for heating and fur maintenance of weak YF mice. The mice were further genotyped according to the protocol from Rockenstein et al. (2002), as mutated tyrosine did not interfere with the sequence used for primer design. All genotyping of mice used for experiments was double-checked.

### 2.2. Behavioral Tests

Behavioral experiments of WT, YF, YF^low1^, YF^low2^, and ASO mice were performed in the light phase, and mice were habituated to the experimental room for at least 1 h. YF mice revealed a strong early phenotype, e.g., seen by their 100% inability to walk on a grid at the age of 1.5 months. The tests, except the force test (see below), were performed at end-point stage (7–8 months), correspondingly to the longest-surviving mice of the litter examined. The behavioral tests were not performed over time in order not to stress the mice through movement of cages and stress upon handling.

#### 2.2.1. Open Field

The open-field test was used to test locomotor ability and anxiety. Mice were placed in a cage with no lid and their activity followed using video tracking software for 20 min. The track paths were compared between genotypes; the test zone was divided into inner, intermediate, and outer zones. The time spent in each zone was determined for each mouse and compared between genotypes. The total distance and individual speed were also recorded and compared. For the experiments shown in Appendix A, an experimental open-field cage was used. The mice were accustomed to the cage the day before the experiments. The experiments shown in Figure 2A were performed with similar settings and measurements in mouse home cages with bedding to further reduce external stress.

#### 2.2.2. Rotarod

The mice were placed on a rotarod (L.S.I. Letica Scientific Instruments Rota-rod LE8200) with increasing speed (linear increase from 4 to 40 RPM in 5 min). The total time able to stay on the rotarod was recorded; each mouse had three trials in the total time measurement for 3 days in a row. The day before testing, the mice were introduced to the rotarod. A Serial Data Communication v 1.4.02 software program was associated with the rotarod to ensure precise tracking.

#### 2.2.3. Elevated Maze

The apparatus consisted of two open arms 5 cm wide and 25 cm long placed across each other, perpendicular to two closed arms 5 cm wide and 25 cm long, with a high closed fence around. The middle contained a center platform (5 × 5 cm). The mice were placed in the center platform and their activity tracked for 15 min with video-tracing record software. Afterwards, the time spent in the different arms was calculated. The maze experiment was considered invalid if the mouse fell of the apparatus.

#### 2.2.4. Object Exploration Test

The set-up from open field was used, but with an unknown object placed in one end of the field. Video-tracking software was used to detect the amount of time taken to explore the unknown object.

#### 2.2.5. Force Test

A force test was performed using a small apparatus to internally compare the ability to hold onto a grid. The mouse was placed on the grid with either the front legs or both front and hind legs and allowed to grab the grid. Then the mouse was pulled away from the apparatus in a horizontal line, until it released its grip. This was repeated three times for each mouse, and an average of the force was calculated. The force was not related to any known measurement of force but only used to compare YF and WT mice internally.

#### 2.2.6. Hind-Limb Clasping Test

Each mouse was held by the tail over the home cage with video tracking with 10 successive trials of 3 s with 3 s rest in between. Each 3 s, the trial was scored as follows: 0: Normal; 1: Hindlimbs with abnormal retraction towards the midline until parallel but without touching; 2: Both hindlimbs retracted with touch or cross (clasp) part of the time; 3: Both hindlimbs clasped within the first second and throughout the time. The score from the 10 trials was summed to 0–30.

#### 2.2.7. Adhesive Removal Test

Small adhesive round stickers, 35 mm in diameter, were placed on the nose of the mouse, and the time to make contact and remove this stimulus recorded. To remove the stimuli, the mouse would raise both forelimbs and swipe away the sticker with the paws. Each mouse had three trials at 60 s each. The average removal time was calculated for each mouse.

#### 2.2.8. Gait Analysis

The mice were first trained to walk on a piece of paper. At the time of the experiment, the paws of the mice were painted with non-toxic paint and the mice placed on a clean piece of paper. The mice walked across the paper; the analysis was considered valid if the mice continued walking. The remaining paw prints were analyzed by measuring the distance between each paw print. The stride length was calculated as the length of the gait, the sway length as the width of the gait, and the stance length as the distance between each hind-limb.

### 2.3. mRNA Extraction and qPCR

RNA was isolated from whole brain of WT (*n* = 8), YF^low1^ (*n* = 5), YF^low2^(*n* = 2), YF (*n* = 2), and ASO (*n* = 3) mice using an RNeasy mini-kit from Qiagen (74,104), following the manufacturer’s instructions. Prior to the extraction, the tissue was homogenized in lysis buffer from the kit using stainless steel beads (Qiagen 69,989) and Tissue LyserII from Qiagen followed by a short spin in a tabletop centrifuge at maximum speed. The RNA quality was checked by the presence of two ribosomal bands on agarose gel. For reverse transcription, 2 µg RNA was made into cDNA using a high-capacity cDNA reverse transcription kit (Applied Biosystems, 4,368,814). Equal amounts of RNA were diluted to the instructed range and 9 μL of sample was mixed with 10 μL of TaqMan Fast Advanced Master Mix and 1 μL of a TaqMan probe; all samples were analyzed in triplicates. Nuclease-free water was used as a negative control and *Nadh* was used as a reference gene. qPCR was performed using a 7500 fast real-time PCR system (Applied Biosystems, Foster City, CA, USA) and quantifications were performed using the 2^-ΔΔCT^-method. For α-synuclein expression, SNCA, a Mm01188700_m1 probe was used.

### 2.4. Western Blotting

Sequential fractionation of brain tissues. To eliminate blood from brain tissue, mice were deeply sedated with subcutaneously injected hypnorm–dormicum and transcardially perfused with PBS with phosphatase inhibitors (25 mM β-glycerolphosphate, 5 mM NaF, 1 mM Na VO_4_). Brains were collected and immediately homogenized in 5 × volume of ice-cold homogenization buffer (320 mM sucrose, 4 mM HEPES-NaOH, 2 mM EDTA, complete protease inhibitor mix (Roche, Manheim, Germany)) containing phosphatase inhibitors by 20 strokes in a loose-fitting glass-homogenizer. For full homogenate samples, a part of the homogenate was subjected to sonication in a cold-water bath using a Branson Sonifier 250. The remaining homogenate was centrifuged at 1000× *g* for 10 min at 4 °C and the pellet discarded. The supernatant was centrifuged at 12,000× *g* for 15 min at 4 °C, revealing the cytosol fraction. The cytosolic fraction used for ELISA was subjected to additional centrifugation at 260,000× *g* for 1 h to ensure that only soluble oligomers were measured by the oligomer ELISA. The pellet from the 12,000× *g* spin was resuspended in 1 mL of homogenization buffer and exposed to another centrifugation of 12,000× *g* for 15 min at 4 °C. The resulting pellet enriched for synaptosomes [16] was extracted with RIPA buffer (50 mM Tris (pH 7,4), 159 mM NaCl, 1% Triton X-100, 2 mM EDTA, 0.5% sodium deoxycholate, 0.1% SDS) for 1 h. Thereafter, samples were spun at 25,000× *g* for 20 min at 4 °C. Protein concentration was measured in all samples using a bicinchoninic acid (BCA) assay.

The samples (full homogenate, cytosol or synaptosomes) were prepared In loading buffer (50 mM Tris, pH = 6.8, 4% SDS, 40% glycerol, bromophenol blue) and 2.5 mM DTT, denatured at 95 °C, run on 8–16% Bis-Tris gels and transferred to PVDF membrane with the iBlot 2 dry blotting system. Afterwards, membranes were fixed in 4% paraformaldehyde (PFA) and boiled for 5 min to detect α-synuclein. Blocking and incubation with primary and secondary antibodies were performed in blocking buffer (5 % skimmed milk, 150 mM NaCl, 20 mM Tris base, 0.05% Tween20, 0.02% azid, including phosphatase inhibitors (25 mM β-glycerolphosphate, 5mM NaF, 1 mM Na_3_VO_4_)). Anti-α-synuclein antibody was made in-house [17]; anti-P129-α-synuclein was from Elan (clone 11A5) and anti-α-tubulin was from Sigma-Aldrich Co, Saint Louis, MO, USA. (T9026). HRP-conjugated immunoglobulin was used as a secondary antibody (Dako, #P0217 and #P0260). A Fuji LAS-3000 Intelligent Dark Box (Fujifilm, Tokyo, Japan) was used to visualize protein bands after treatment with ECL. A PageRuler Prestained Protein ladder 10–180 kDa (Thermo Fisher, Waltham, MA, USA) was used to estimate the molecular weight of the proteins. Protein bands used for comparison were always present on the same membrane; they were loaded in equal amounts and compared to the reference gene α-tubulin. Quantifications of western blots were performed in ImageJ.

### 2.5. Immunohistochemistry

The mice were euthanized with an overdose of subcutaneously injected hypnorm–dormicum and transcardially perfused via the left ventricle, first with saline and then 4% paraformaldehyde (PFA) in 0.01 M potassium-phosphate-buffered saline (KPBS), pH 7.4. The brains and spinal cords were dissected and post-fixed in 4% PFA overnight at 4 °C. Serial coronal sections (40 μm) were cut on a cryostat and collected free-floating in 0.1 M KPBS, pH 7.4, in sequential series of six. The sections were subjected to immunohistochemistry [18]. In brief, sections of brains and spinal cords were pre-incubated in blocking buffer, consisting of 3% swine serum diluted in 0.01 M KPBS with 0.3% Triton X-100 (Sigma) for 30 min at room temperature to block any unspecific binding. Alternate sections intended for immunolabeling of mouse primary antibodies were blocked for non-specific background using a Mouse-on-Mouse (M.O.M) Basic kit (Vector, Newark, CA, USA). The sections were then incubated overnight at 4 °C with one of the following primary antibodies diluted in blocking buffer: polyclonal anti-rabbit full length α-synuclein (ASY6) [17]) 1:500, monoclonal anti-rabbit α-synuclein (MJFR1, Abcam, Cambridge, UK) 1:5000, monoclonal anti-rabbit α-synuclein (MJF 14-6-4-2, Abcam, Cambridge, UK) 1:5000, polyclonal anti-rabbit phosphorylated α-synuclein [17] 1:1000, polyclonal anti-rabbit glial fibrillary acidic protein (GFAP) (Dako, Glostrup, DK) 1:200 for astrocytes, or monoclonal rat anti-mouse IBA1 (Serotec, Kidlington, UK) 1:5000 for microglia. Next day, the sections were incubated for 30 min at room temperature with biotinylated swine anti-rabbit immunoglobulin (Dako, Glostrup, DK) or biotinylated goat anti-mouse immunoglobulin (Dako, Glostrup, DK) or biotinylated goat anti-rat immunoglobulin (Dako, Glostrup, DK), all diluted 1:200 in KPBS. Binding of the antibodies was visualized using the ABC-system (Vector, Newark, CA, USA) and 3,3′-diaminobenzidine tetrahydrochloride (DAB).

### 2.6. Quantification of Aggregate Size

The size of the ASY6 immunolabelled aggregates in YF (age 5–8 months, *n* = 3) and ASO (age 5–8 months *n* = 3) mice were quantified. Two brain regions, globus pallidus (GP) and pyramis (PY), were imaged using a Zeiss AxioCam MRc coupled to a Zeiss Axioplan 2 imaging microscope. The identification of each section was randomized and a blinded quantification of the area (µm^2^) and diameter (µm) of the six largest aggregates in each image was performed using Fiji Is Just ImageJ (Fiji) [19].

### 2.7. Oligomer/Total α-Synuclein ELISA

Oligomer ELISA was performed as previously described by Lassen et al., 2018 [20]. In brief, 96-well Maxisorp plates were incubated with the antibody MJF-14-6-4-2 that recognizes oligomeric α-synuclein or ASY-1 [17] for measurement of total α-synuclein levels. After blocking with 10% fetal calf serum (FCS) in PBS, plates were incubated with equal amounts of brain homogenates from YF, YF^low1^, YF^low2^, WT or ASO mice overnight, followed by treatment with secondary (anti-α-synuclein, BD 610787) and tertiary antibody (anti-mouse HRP). The amounts of oligomer or total α-synuclein were determined by color detection at 450 nm after the addition of 3,3′,5,5-tetramethylbenzidine (TMB), and subsequently 1M phosphoric acid, to terminate enzymatic activity.

### 2.8. Microtubule Retraction Assay

The oligodendroglial immortalized cell line (OLN-93 (WT)) was derived from glial cells obtained from brains of Wistar rats [21]. Cells were cultured at 37 °C, 5% CO_2_ in Dulbecco’s Modified Eagle’s 46 Medium (DMEM), (Lonza, Basel, Switzerland) supplemented with 10% FCS and 50 μg/mL penicillin/streptomycin. The OLN-93 model of α-synuclein aggregate stress depends on the co-expression of α-synuclein and the protein p25a/TPPP that stimulates aggregation of α-synuclein [22]. Plasmids were prepared by inserting the coding sequence of the human α-synuclein gene into the pcDNA3.1/zeo(-) vector (Invitrogen, Waltham, MA, USA) at XbaI and NotI restriction sites. α-synuclein was synthesized with mutations at S129 and/or Y125, Y133, and Y136 and inserted stepwise to generate α-synuclein SA, α-synYF and αsynSAYF vectors, respectively. The sequences of the vectors were verified by sequencing at Eurofins MWG Operon using standard CMW primers. Transient transfections were performed using FugeneTM 6 transfection reagent (Roche, Manheim, Germany) according to the manufacturer’s instructions using 0.25 μg of plasmid DNA for 12-well plates. Transfections were carried out in DMEM + 0.05% FCS.

Transfected OLN-93 WT cells were cultured on poly-L-lysine coated coverslips for 24 h and fixed in 4% PFA in PBS for 10 min, permeabilized in permeabilization buffer (50 mM glycine, 0.1% Triton-X-100, PBS), and blocked with 3% BSA in PBS. The cells were then incubated with primary and secondary antibodies and subsequently mounted with mounting medium containing DAPI (Dako, Glostrup, DK). Microtubule (MT) retraction [23] was determined on a Zeiss Axiovert 200 M inverted fluorescence microscope and quantified as the percentage of α-synuclein and p25 positive cells with MT retraction.

### 2.9. Statistics

Data is depicted as mean ± standard deviation (SD); a *p*-value < 0.05 was considered statistically significant. All graphs and statistical analyses were produced using GraphPad Prism (version 9.3.1), except Figure 8 which was produced in Excel. Since the experimental groups varied in size, and most contained less than six samples, no test for normality was included. Instead, datasets were analyzed for equal variances between the groups (WT, YF, YF^low1^, YF^low2^, and ASO) using an F-test or Brown–Forsythe test, depending on the number of groups. If the variances were equal, a parametric unpaired *t*-test or one-way ANOVA followed by Sidak post hoc test for multiple comparisons was performed; datasets that had significantly different variances were analyzed using a non-parametric Mann–Whitney test or Kruskal–Wallis test followed by Dunn’s multiple comparisons test. In the microtubule retraction assay, at least 100 cells were counted for each experiment; the experiment was repeated three times.

## 3. Results

### 3.1. YF Mice Express α-Synuclein at Lower Levels Than ASO Mice

To study the effects of tyrosine phosphorylation in vivo, a transgenic mouse line YF (original name: *TgN(Thy-1-SNCA-YF)4Emfu*) was made by pronucleus injection of α-synuclein with three tyrosine residues in positions 125, 133, and 136 mutated to phenylalanine. Transgenic YF α-synuclein was inserted downstream to the thy1-promoter with the vector used to produce the well-characterized and well-known overexpression model of α-synuclein, the ASO mouse, also known as “line 61” [15] (Figure 1A). The ASO mouse line was used as control animals to make sure that the qualitative effects of tyrosine mutations were studied as well as the quantitative effects of α-synuclein overexpression alone. We aimed to obtain a similar or slightly smaller expression level than with the ASO mouse; the amount of expression in the pronucleus injection models was decided by the amount of vector copies inserted as well as at the genomic insertion site.

As we expected, the YF mouse line exhibited an expression level that was slightly less than for the ASO mouse, therefore ensuring that differences were caused by the YF mutations and not by the level of α-synuclein overexpression (Figure 1B). The level of expression observed with Western blotting was confirmed by RT-qPCR analysis showing lower gene expression of *Snca* in YF mice compared to ASO mice (Figure 1C). Although early behavioral changes with α-synuclein aggregate deposition can be found in the ASO model, phenotypic expression is modest, as ASO mice have a normal lifespan and do not exhibit extensive α-synuclein pathology [15,24,25]. In contrast, the YF mice displayed a severe phenotype starting at one month of age and often had a decreased lifespan (Figure 1D). The mice were dramatically impaired in terms of motor performance, as seen by the part paralysis of both hind- and forelegs, as well as being significantly smaller (Figure 1E). The YF mice were unable to breed and had to be produced through in vitro fertilization. Some mice survived up until eight months, whereas others were so impaired in terms of motor function that they were sacrificed after 1.5 months (Figure 1D). The YF mice were able to gain weight during their lifespan despite their smaller size and dramatic phenotype (Figure 1F). Due to the inability of the mice to breed, the mice were not genetically backcrossed, which might explain their different capability to cope with the increased expression of YF α-synuclein. Due to the inability to breed, the mice were kept heterozygous, significantly reducing the risk that the phenotype was a secondary effect of the insertion site within the genome rather than an effect of YF α-synuclein overexpression.

We extended our analysis and generated two other YF α-synuclein mouse lines. The first line YF^low1^ (original name: *TgN(Thy-1-SNCA-YF)2Emfu*) had a low amount of YF α-synuclein overexpression, which was close to the amount of normal α-synuclein found in a WT mouse (Figure 1B,C). The other line, YF^low2^ (official name: *TgN(Thy-1-SNCA-YF)1Emfu*), had a slightly higher overexpression than YF^low1^, as evidenced by Western blotting (Figure 1B). RT-qPCR analysis confirmed a higher expression level of YF^low2^ compared to YF^low1^ (Figure 1C). Both YF^low^ models did not show any obvious signs of motoric disabilities (see below), and, in terms of the cerebral distribution of α-synuclein detected using the ASY6 antibody, the two YF^low^ strains were indistinguishable from the WT mouse (Appendix A).

### 3.2. YF Mice Are Severely Motor-Impaired

Examining the motor capability of YF mice, we initially found that these had difficulty performing motor tests due to their severe phenotype, e.g., the pole and hanging grip tests were completely impossible for the mice to perform, while the YF mice were unable to walk in the open arms of the elevated plus maze test (Figure 2; Appendix A). However, it was possible to perform an open-field analysis of all mice strains, which we undertook in two different settings. In a normal open-field test setting (performed at 7.5 months of age), YF mice showed a significantly reduced level of activity compared to WT and ASO mice (Appendix A). WT mice spent most time in the intermediate and peripheral zones of the open-field cage, and, despite some hyperactivity, ASO mice also preferentially spent most time in the outer zones. To avoid external stress and inability to walk on a slippery surface, we also performed open-field testing of the YF mice in a normal mouse cage and with bedding within the cage, and compared the results to those for WT and ASO mice (Figure 2). Again, YF mice showed reduced activity (Figure 2A) and speed (Figure 2B) compared to WT mice at the same level as seen for the open-field test performed in the open-field test cage. The time spent in different zones of the cages in terms of percentage was equal to that of WT and ASO mice, with an increased amount of time spent in intermediate and peripheral zones. A clasping test performed on YF mice (Figure 2D) (7.5 months) compared to WT and ASO mice showed that YF mice exhibited extensive clasping at all times, reflected in a full score of 30 (scoring system described in Materials and Methods section) for all YF mice tested. WT mice did not clasp at all, while ASO mice showed a small amount of clasping at this age, similar to the level observed in Sampson et al., 2016 [24] (not shown). In the adhesive removal test, the YF mice showed obvious signs of will to remove the sticker; however, their motoric impairments resulted in complete inability to remove stickers; all mice spent 60 s trying, which was the maximum time tested (Figure 2E). ASO mice are reported in the literature to also have reduced ability to remove the sticker, albeit not at a level comparable to the YF mice [24]. We also found that ASO mice were able to remove the sticker, but they did this less quickly (not shown), probably due to induced stress after handling, in contrast to the YF mice which were unable to remove the sticker at all. Force measurements (Figure 2F) showed that YF mice had no strength to pull a grid, so the motor impairment was due not only to balance problems but also to reduced strength.

Gait analysis (Appendix A) revealed that the YF mice had reduced stride and sway length, as has also been reported in other mice models of Parkinson’s disease, which probably reflects basal ganglia dysfunction and/or dopamine dysregulation [26,27]. This is in accordance with the observed accumulation of aggregates containing α-synuclein along the pyramidal tract (see below).

To study the effects of the low level of overexpression of YF α-synuclein, the YF^low^ mice were also subjected to behavioral analyses. YF^low1^ strain mice were tested on the rotarod at the age of nine months, where they performed as well as WT control littermates (Appendix A). Open-field analysis of YF^low1^ and WT at both 7- and 12–14 months of age did not reveal any differences either (data from 12–14 months shown). The distance travelled was equal for YF^low1^ and WT mice and the amount of time spent in the different zones of the open-field area was also equal (Appendix A). Interestingly, when the YF^low1^ mice were tested in the elevated plus maze test, they spent significantly more time within the open arms compared to WT mice (Appendix A). The distance travelled within open arms was also increased for YF^low1^ mice compared to WT mice, showing that YF^low1^ mice not only sat in the open arms but also explored the field, which was indicative of reduced fear experience (Appendix A). The data shown in S1 show the first test results of each individual mouse in a group of 13 YF^low1^ and 8 WT mice. To test the validity of the results, the mice were tested three times in total, with equal results obtained in each instance. To confirm the observation of reduced anxiety behavior, we analyzed the behavior of the YF^low2^ line, where we observed the same tendency (not shown). To test the exploration of an unknown object, we recorded the exploration time for two objects in an open-field area by YF^low1^ mice; we found that they spent more time exploring the objects compared to WT mice, supporting the inference of reduced anxiety in this mouse model (Appendix A). YF^low2^ mice showed modest clasping or abnormal hindlimb behavior when they aged, whereas YF^low1^ mice, even in the oldest age range, did not clasp, nor were any other motor disabilities observed.

### 3.3. YF Mice Accumulate α-Synuclein in the Synaptosomes

By fractionation of brain homogenates from YF and ASO mice and comparing the relative amounts between different cell compartment between ASO and YF, we first confirmed, by Western blotting, that the total amount of α-synuclein was slightly lower in YF mice compared to ASO mice (Figure 3A). The amount of α-synuclein in the cytosol fraction was also slightly lower for YF mice compared to ASO mice (Figure 3B). In contrast, the amount of α-synuclein within the synaptosomal fraction was higher for YF mice compared to ASO (Figure 3C), indicating that tyrosine-deprived α-synuclein tended to accumulate within the synapses, or that the threshold for transport of α-synuclein to the synapse was related to the phosphorylation status of the protein.

To study the latter in more detail, we examined the phosphorylation status of serine 129 within the cytosol and synaptosomes of the brain. Based on previous reports [8,9], we anticipated that the amount of serine phosphorylation would be decreased upon removal of tyrosine phosphorylation. A slight but significant increase in the phosphorylation level of serine 129 in the cytosol was seen in the YF mice compared to ASO mice (Figure 3B). In contrast, a significantly decreased phosphorylation level of serine 129 was seen in the synaptosome fraction for YF mice compared to ASO mice (Figure 3C)—in both cases the phosphorylation level of serine 129 was relative to the internal α-synuclein level. To study whether the decrease in serine phosphorylated α-synuclein within the synaptosomal fraction related to high levels of α-synuclein in YF mice, we examined the low-expressing model YF^low1^. We found increased quantities of α-synuclein in the synaptosome and that the amount of serine 129 phosphorylated α-synuclein compared to total α-synuclein was significantly lower in the YF^low1^ model compared to the WT model (Figure 3D), as we also saw for the YF mice. This supports our hypotheses that the ability to phosphorylate tyrosine is important for serine phosphorylation, cytosolic transport, and deposition of serine phosphorylated α-synuclein in the synapse, even in YF^low1^ mice expressing close-to-normal levels of α-synuclein.

### 3.4. YF Mice Have High Amounts of Soluble α-Synuclein Oligomers

α-synuclein oligomers are believed to be highly toxic [28]. As the distribution of α-synuclein was indistinguishable between YF and ASO mice concerning the aggregation specific antibody MJF-14-6-4-2, we sought to evaluate the oligomer level in YF mice using a quantitative assay. We previously designed a special oligomer ELISA [20] lacking the capability to detect monomeric α-synuclein, demonstrated by the fact that addition of high amounts of recombinant monomeric α-synuclein to WT mouse brain samples did not reveal any false positive signal. Using the oligomer ELISA assay, it was evident that YF mice have significantly higher amounts of soluble oligomers compared to age-matched ASO mice in both the cytosol and synaptosomes (Figure 4A,B). Moreover, a small significant increase was also found in YF^low1^ and YF^low2^ mice compared to WT mice, indicating that YF α-synuclein is more prone to form toxic oligomers even at low levels (Figure 4C).

### 3.5. YF Have High Amounts of α-Synuclein in Cerebral Cortex and Limbic Structures

As the toxic effect of YF α-synuclein was not caused by aberrantly high levels of serine 129 phosphorylated α-synuclein, we examined age-matched YF, ASO, and WT mice for α-synuclein depositions by histological examination. YF mice deposited full-length α-synuclein, as detected by antibody ASY6, in neurons widespread in the CNS, with a main occurrence in forebrain regions, such as the cerebral cortex and basal ganglia (Figure 5, Appendix A), and limbic structures, such as the hippocampus, septum, and amygdala (Appendix A). Labeling was also prominent in many nuclear regions of the brain stem, deep cerebellar nuclei, and cerebellar cortex. The expression in the cerebral cortex was profound in neurons of all cortical layers. From neurons of motor cortex layer V, forming the pyramidal tract, it was evident that the axons contained α-synuclein aggregate-like structures extending to their terminals in the dorso-lateral portion of the spinal cord (Appendix A). On route to the spinal cord, the axons of the pyramidal tract contained α-synuclein-containing aggregate-like structures in the internal capsule, crus cerebri, and pyramis (Figure 5B). We observed early-onset (1.5 month of age) accumulation, but the young mice had lower accumulation of α-synuclein-containing aggregates than old (aged 8 months) YF mice (Figure 5). The protrusion of the accumulation of the α-synuclein aggregates tended to progress in a proximo-distal direction as aggregate-like structures and was hardly seen in spinal cords at the lumbo-sacral level. A gradually increasing presence of α-synuclein aggregate-like structures in extra-pyramidal regions was also evident. Changes were slight, yet clearly present, in the striatum (Figure 5). Other brain circuits related to impulse trafficking through the basal ganglia, i.e., the globus pallidus, entopeduncular nucleus, and substantia nigra pars reticulate, were prominent with respect to accumulation of aggregate-like structures (Appendix A).

The brains and spinal cords were also examined for their distribution of phosphorylated α-synuclein to further address the state of activation of α-synuclein at the morphological level. The phosphorylation of α-synuclein was generally most prominent in neurons of the forebrain, which can be attributed to the transgenic expression of mutated α-synuclein, driven by the Thy-1 promotor, which is mainly expressed in the forebrain. The phosphorylated α-synuclein was observed in neurons in both the nucleus and perinuclear cytosol. Moreover, when examined at older ages, it was evident that YF mice also had phosphorylated α-synuclein expression protruding into peripheral processes with proximal dendrites being especially evident (Figure 6).

For a comparison of the aggregate size and diameter between YF and ASO mice, we quantitatively analyzed sections stained for full-length α-synuclein using the ASY6 antibody. These analyses revealed a profound difference between these two mice strains with the YF mice showing clearly larger aggregates in terms of aggregate size and diameter (Figure 7).

The probable deleterious effects of mutating the three tyrosines within the C-terminal part of α-synuclein were also studied in the oligodendrocyte cell line (OLN-93), where overexpression of α-synuclein and p25α together are known to lead to cell toxicity, as measured by microtubule retraction [22]. As previously demonstrated by Kragh et al., 2009 [22], mutation of serine 129 to alanine led to decreased toxicity, showing that α-synuclein toxicity was dependent on phosphorylation of serine 129 (Figure 8). Interestingly, mutations of three tyrosines, 125, 133, and 136, counteracted this decrease in toxicity. Mutations of tyrosines in α-synuclein carrying α-synuclein with serine 129 mutated to alanine did not induce toxicity. This suggests that the three tyrosines, 125, 133, and 136, possess a permissive role for α-synuclein pS129-dependent aggregate toxicity but do not on their own induce toxicity in this model.

## 4. Discussion

The results presented in this study show that insertion of three tyrosine residues of the α-synuclein gene in positions 125, 133, and 136 mutated to phenylalanine led to a severe phenotype. Based on previous observations demonstrating that serine 129 phosphorylation of α-synuclein depends on phosphorylation of tyrosine 125 as an initiating event [8,9], we hypothesized that the post-translational modification of α-synuclein caused by insertion of three tyrosines would lead to serine 129 phosphorylation, which was previously shown to cause neurotoxicity in vitro and in *Drosophila* [12,29]. That the ability to phosphorylate tyrosine is important for serine phosphorylation and cytosolic transport of α-synuclein was evident, as it was observed that, in the synapses of YF^low1^ mice expressing close-to-normal levels of α-synuclein, the amount of serine 129 phosphorylated α-synuclein was significantly lower compared to that found in ASO mice in response to the lowering of tyrosine phosphorylation. However, as serine 129 phosphorylation of α-synuclein in YF mice, somewhat surprisingly, decreased upon tyrosine substitution compared to their ASO origination, we found that the pathology caused by tyrosine mutation was not caused by serine 129 phosphorylation of α-synuclein. Nonetheless, we were able to detect a distinctively higher level of phosphorylation in neuronal nuclei and the perinuclear cytosol of the YF mouse. This observation was not higher in comparison with the ASO line but indicates that the alteration of α-synuclein phosphorylation in genetic models leads to effects which are consistent with those seen in models with expression of mutant forms of α-synuclein.

The transgenic YF mouse line exhibited an expression level of α-synuclein that was slightly lower than for the original human α-synuclein transgenic ASO mouse, as verified at both the protein and mRNA levels, indicating that the differences between the YF and ASO mouse lines were caused by the YF mutations and not by the level of α-synuclein overexpression. The earlier initiation of clinically aberrant features and short lifespan seen in YF mice also emphasizes the uniqueness of the phenylalanine substitutions of the α-synuclein gene for causing pathology. Underlining the difference between the YF and ASO mouse lines, the YF mice showed significantly larger aggregates. The observations made in this study may add to the understanding of early-onset Parkinson’s disease, particularly in cases with a mutation of α-synuclein leading to tyrosine replacement with phenylalanine. The clinical aberrations observed in the YF mice were closely correlated with the pathological deposition of α-synuclein aggregates. α-synuclein aggregates were seen in telencephalic regions, including the motor cortex, hippocampus, and limbic regions, such as the septum and amygdala, which are regions that play an important part in the neuronal circuits involved in the tasks examined in the behavioral studies. The formation of α-synuclein aggregates is indicative of neuropathology—we are convinced that the deposition of such aggregates, e.g., in the motor cortex and corticospinal tract, involving pyramidal neurons of the motor cortex and spinal cord and involvement of extrapyramidal pathways in the striatum and substantia nigra, together contributed significantly to the clinical symptoms observed in the behavioral tests. We also looked for other signs of neuropathology, such as activation of glial cells, but this was not observed in the immunohistochemical stains for reactive GFAP positive astrocytes or IBA-1 positive microglia (unpublished observation), implying that neuroinflammation was occurring slowly, which is also observed in Parkinson’s disease autopsies [30].

The reduced anxiety behavior observed for YF^low^ mice was also seen for other synuclein transgenic mice, as both heterozygous M83 A53Tα-synuclein mice under the prion promoter [31] and ASO mice exhibited reduced anxiety in the elevated plus maze (not shown). The level of α-synuclein overexpression in these models was significantly higher than in the YF^low^ models and suggests that only a small amount of α-synuclein in the tyrosine-mutated α-synuclein is needed to cause the change in behavior in mice, without affecting the motoric ability of the mice.

A difference observed between the fractionates of homogenates of YF and ASO mice was the higher accumulation of α-synuclein within synaptosomes in the YF mice. This indicates that the tyrosine-deprived α-synuclein of the YF line tended to accumulate within the synapses, or that the threshold for transport of α-synuclein to the synapse was related to the phosphorylation status of the protein. The observations from the synaptosome fractions imply that non-phosphorylated (both tyrosine and serine) α-synuclein accumulated in the synapse, which was also evident from observation using immunohistochemistry, which showed both normal and phosphorylated α-synuclein in the synapse. The data obtained also suggest that the ability to phosphorylate tyrosine increases the amount of serine 129 phosphorylated α-synuclein in the synapse, which is consistent with previous observations [32]. The implications of the latter await further investigation but suggest that the extent of phosphorylation of α-synuclein influences the balance between anterograde and retrograde axonal transport, leading to increased net transport of α-synuclein from the neuronal soma to the axonal terminal.

In cultured oligodendrocytes overexpressing α-synuclein, which are prone to intoxication, mutation of serine 129 led to decreased toxicity. This was also observed in the oligodendrocytes expressing tyrosine of the α-synuclein gene in positions 125, 133, and 136. The current study did not explore the possible neuroprotective role of phosphorylation of α-synuclein but suggests that this may be the case, as judged from the observations made in the cultured oligodendrocytes.

In cultured oligodendrocytes, co-expression of α-synuclein and p25a exhibit an aggregate toxicity relying on the phosphorylation of serine 129 [22]. We demonstrate here that this serine-129-dependent cytotoxicity was dependent on tyrosines 125, 133, and 136 being intact, as their mutation to phenylalanine abrogated the serine-129-dependent toxicity. Surprisingly, the α-synuclein protein with the tyrosines mutated, but with serine 129 intact, remained toxic. This suggests that the tyrosines, potentially due to their phosphorylation, are not toxic on their own, but that they facilitate the toxicity mediated by phosphorylated serine 129. The current study did not explore the possible neuroprotective potential of modulating serine and tyrosine phosphorylation in the C-terminus of α-synuclein, but, collectively, our in vivo and ex vivo data support a potential role in the modulation of α-synuclein-dependent neurotoxicity.

## Figures and Tables

**Figure 1 cells-11-03673-f001:**
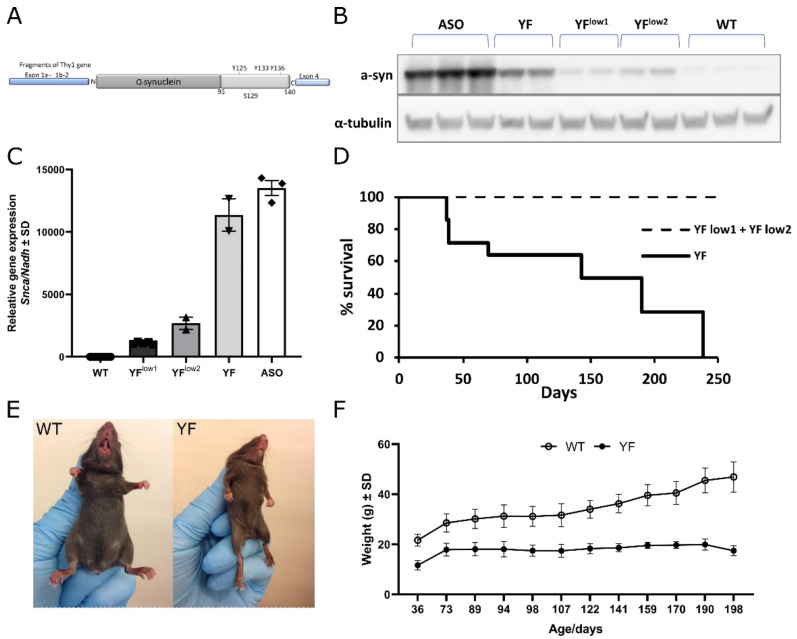
**Lack of tyrosine phosphorylation is toxic in a mouse model.** (**A**). Generation of human α-synuclein YF transgenic mouse lines by insertion of α-synuclein with three mutated tyrosines (125, 133, and 136) for phenylalanine downstream of the Thy1 promoter. The DNA was inserted into the mouse genome by pronucleus injection. YF mouse lines were generated with the mThy1.2 cassette as the ASO mouse line [15]. (**B**). Western Blotting of whole brain homogenates from ASO mice, mice of the three YF mouse lines, and wildtype (WT) mice. The YF mice have a slightly lower expression level of α-synuclein compared to that of ASO mice. YF^low1^ has a slightly lower expression than YF^low2^ (**C**). Relative gene expression of the *Snca* RNA level in WT, the three different YF mouse lines, and ASO mice compared to the reference gene *Nadh*. The profile of the expressed RNA levels corresponds to protein levels demonstrated in B. Data are depicted as mean ± SD: WT (*n* = 8), YF^low1^ (*n* = 5), YF^low2^ (*n* = 2), YF (*n* = 2), and ASO (*n* = 3). (**D**). Kaplan–Meyer curve of a cohort of YF mice (*n* = 14) compared to YF^low1^ and YF^low2^ (both *n* = 14). YF mice have decreased life span, whereas YF^low1^ and YF^low2^ mice have normal life span, even at time points beyond 250 days. YF mice were sacrificed at humane end-points due to motoric difficulties starting at 45 days of age. (**E**). Images of adult WT and YF mice. YF mice are smaller and have a dramatic motoric phenotype seen by paralysis of both hind- and forelegs starting at one month of age. (**F**). YF mice (*n* = 7) are significantly smaller than their WT littermates (*n* = 10) during the entire experimental period, *p* < 0.005 for all time points. Nonetheless the YF mice gain some weight. Data are depicted as mean ± standard deviation (SD); as variance was equal for the two experimental groups, the data was analyzed using unpaired *t*-test.

**Figure 2 cells-11-03673-f002:**
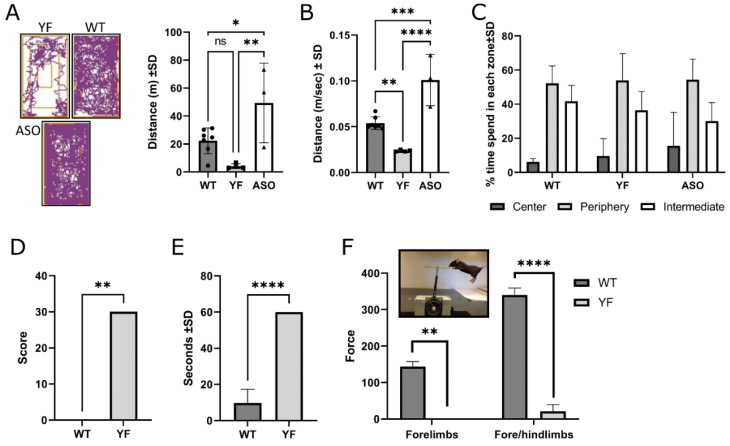
**Behavioral compromises in YF mice.** (**A**). Open-field analysis shows reduced walking distance by YF mice (7.5 months; *n* = 4) compared to wildtype (WT; *n* = 7) and ASO mice (*n* = 3). (**B**). In the open-field test, YF mice also exhibit reduced walking compared to WT and ASO mice (speed measured in the mobile time (m/s)). (**C**). YF mice spend similar amounts of time in center, periphery, and intermediate zones compared to WT and ASO. (**D**). Clasping measured in YF and WT mice. Full clasping with both hind legs gives a score of 3. All YF mice (*n* = 4, 7.5 months) received a full score of 3 in 10 tests (30 in total). WT mice (*n* = 7) do not clasp at all resulting in a final clasping score of 0. (**E**). In the adhesive removal test, YF mice (7.5 months; *n* = 4,) are not able to remove the sticker placed on their noses during the experimental time of one minute. Each YF mouse is allowed to attempt to remove the sticker 3 times for 1 min each time. WT mice (*n* = 6) are all able to remove the sticker when tested 1 or 2 times. (**F**). Force measured by a small apparatus (insert) measuring the mouse’s ability to hold on to a grid. WT mice (*n* = 8) pull with a strength of 144 with their forelimbs, whereas YF mice (*n* = 4) are unable to pull the grid. With all four legs, the WT mice pull with a strength of 340 (close to maximum limit of apparatus), whereas YF mice pull with a force of only 21. Data are depicted as mean ± standard deviation (SD); depending on the variance of the data and the number of experimental groups, the data was analyzed by one-way ANOVA with multiple comparisons, Mann–Whitney test or unpaired *t*-test. * *p* < 0.05, ** *p*  <  0.01, *** *p*  <  0.001, **** *p* < 0.0001.

**Figure 3 cells-11-03673-f003:**
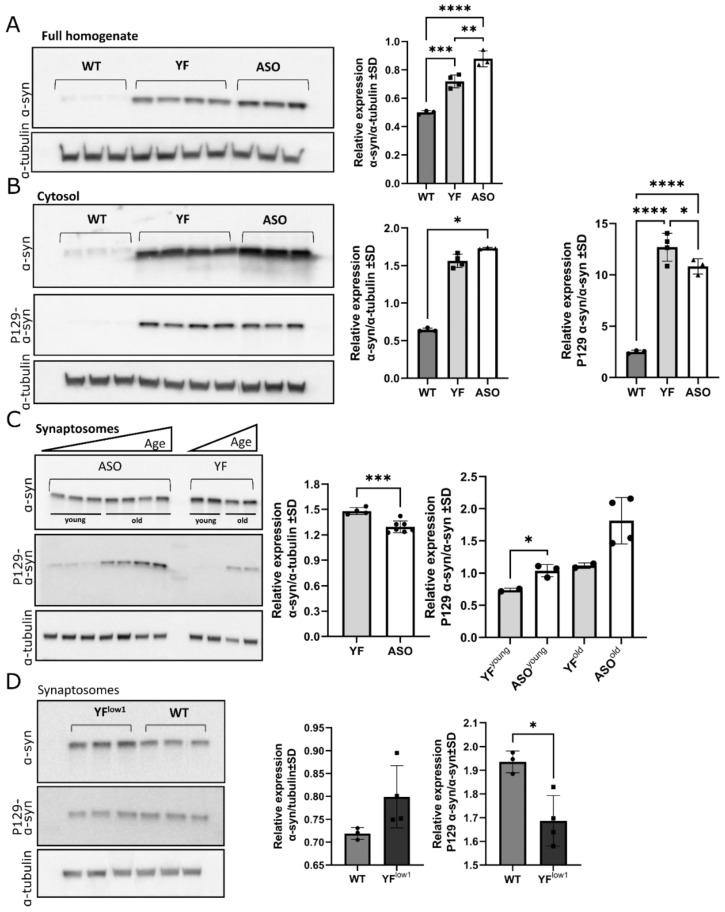
**Biochemical determination of P129 phosphorylation of α-synuclein (α-syn) in wildtype (WT), YF, and ASO mice.** (**A**). Expression of α-synuclein and α-tubulin in whole brain homogenates (**left**) and the quantification of α-synuclein relative to α-tubulin (**right**). The YF mice (*n* = 4) express significantly more α-synuclein compared to WT (*n* = 3) and less when compared to the ASO mice (*n* = 3). (**B**). Expression of α-synuclein, P129 α-synuclein, and α-tubulin in the soluble part of the brain homogenate (cytosol) (**left**), and quantification of α-synuclein relative to α-tubulin (**middle**) and P129 α-synuclein relative to α-synuclein (**right**). Both YF (*n* = 4) and ASO (*n* = 3) have increased expression of α-synuclein in the cytosol compared to the WT (*n* = 3), and, although not significant, α-synuclein levels seem lower in YF compared to ASO mice. The amount of P129 α-synuclein is significantly increased in YF both compared to WT and ASO). (**C**). Expression of α-synuclein, P129 α-synuclein, and α-tubulin in brain samples enriched for synaptosomes (**left**), and the quantification of α-synuclein relative to α-tubulin (**middle**) and P129 α-synuclein relative to α-synuclein (**right**). Significantly increased α-synuclein expression is seen in YF mice (*n* = 4) relative to ASO (*n* = 7) and WT mice (*n* = 3), showing that YF have increased α-synuclein in the synaptosomes. P129 α-synuclein increases with age in both YF and ASO mice, but P129 α-synuclein levels in YF mice are decreased compared to ASO in both young and old mice (**right**). ASO mice are examined at 1.5–4 months (young) and 7–8 months (old), respectively. YF mice are 4- (young) and 8-months-old, respectively. (**D**). Expression of α-synuclein, P129 α-synuclein, and α-tubulin in brain samples enriched for synaptosomes in YF^low1^ mice (*n* = 3) and WT mice (*n* = 3) (**left**) and quantification of α-synuclein relative to α-tubulin (**middle**) and P129 α-synuclein relative to α-synuclein (**right**). The level of α-synuclein is increased in YF^low1^ mice compared to WT (**middle**), whereas the level of P129 α-synuclein is significantly lower in YF^low1^ mice compared to WT (**right**). Data are depicted as mean ± standard deviation (SD); depending on variance of the data and number of experimental groups, the data was analyzed with one-way ANOVA with multiple comparisons, Kruskal–Wallis with Dunn’s multiple comparisons test, or unpaired t-test, * *p* < 0.05, ** *p* <  0.01, *** *p*  <  0.001, **** *p* < 0.0001.

**Figure 4 cells-11-03673-f004:**
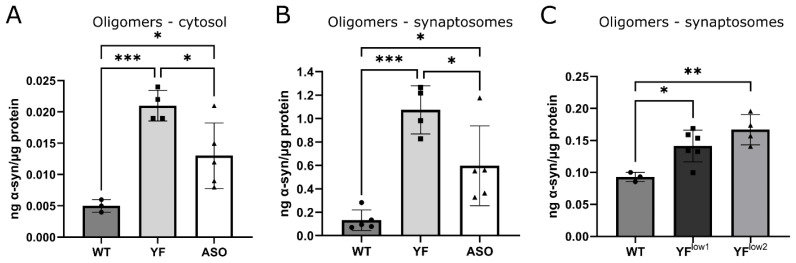
**α-synuclein oligomer levels are increased in YF mice.** (**A**). Quantitation of oligomers measured by ELISA showing that YF mice (*n* = 4) have significant higher amounts of soluble oligomers compared to age-matched ASO (*n* = 5) and wildtype (WT) mice (*n* = 3) in the cytosol. (**B**). The concentration of soluble oligomers in RIPA extracts of synaptosomes is significantly higher in YF mice (*n* = 4) compared to age-matched ASO (*n* = 5) and WT (*n* = 5) mice. All mice in A and B are 4–8 months. (**C**). Concentration of oligomers in the synaptosomes is increased in both 20-month-old YF^low1^ (*n* = 6) and YF^low2^ (*n* = 4) mice compared to WT mice (*n* = 3). Data are depicted as mean ± standard deviation (SD); the variance did not differ between experimental groups and, thus, data was analyzed with one-way ANOVA with a multiple comparisons test. * *p* < 0.05, ** *p*  < 0 .01, *** *p*  <  0.001.

**Figure 5 cells-11-03673-f005:**
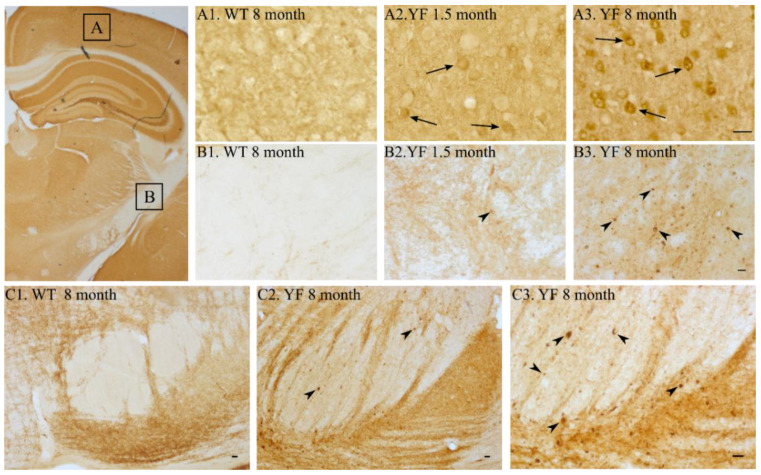
**Distribution of full-length α-synuclein in wildtype (WT) and YF mice in cerebral cortex and pyramidal tract.** The level of α-synuclein in the cerebral cortex (A) and internal capsule (B) is shown in the upper two rows and boxed in the illustration shown to the right using an antibody against full-length α-synuclein (ASY6). (**A1**,**B1**). WT at 8 months. (**A2**,**B2**). YF α-synuclein mouse at 1.5 months. (**A3**,**B3**). YF α-synuclein mouse at 8 months. (**A1**–**A3**). Neurons of the cerebral motor cortex. (**A2**,**3**). Cortical neurons increase their level of α-synuclein with increasing age. (**B1**–**B3**). The internal capsule. Aggregates (arrowheads) are seen in axons of YF α-synuclein mice with increasing age. (**C1**–**C3**). The pyramis at the pontine level. Scale bars: (**A1**–**A3**), 20 μm (bar shown in **A3**), (**B1**–**B3**), 20 μm (bar shown in **B3**), (**C1**–**C3**), 50 μm (bar shown in **C3**).

**Figure 6 cells-11-03673-f006:**
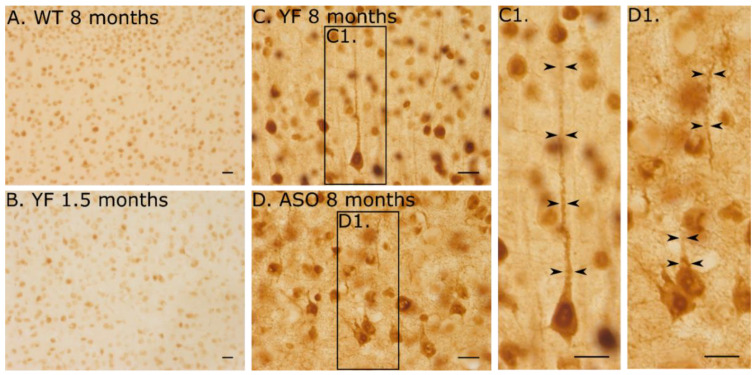
**Phosphorylation in neocortical neurons of wildtype (WT), YF, and ASO brains**. Immunolabelling of phosphorylated α-synuclein in cerebral cortex of (**A**). WT mice at 8 months. (**B**). YF at 1,5 months, (**C**–**C1**). YF mice at 8 months, and (**D**–**D1**). ASO mice at 8 months. Phosphorylated α-synuclein distributes to neuronal perikarya and neurons throughout the cerebral cortex even in the WT mouse. By 1.5 months, neurons are seen with a marked increase in labelling of neuronal nuclei and surrounding cytoplasm (**B**). (**C**) At 8 months, phosphorylation of α-synuclein extends widely into the dendrites (**C**), which is clearly observed at high-power magnification ((**C1**) larger magnification of square in (**C**). (**D**) ASO mice also have increased phosphorylation of α-synuclein, which extends into proximal arborizations of the cytoplasm, though not to the same extent as the YF mice. (**D1**) is larger magnification of square in (**D**). Scale bars: (**A**,**B**), 50 μm. (**C**), 30 μm, (**C1**), 10 μm, (**D**), 30 μm, (**D1**), 10 μm.

**Figure 7 cells-11-03673-f007:**
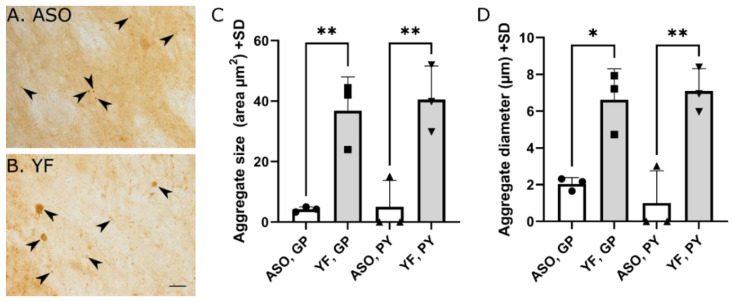
**Quantification of aggregates in 8-month-old ASO and YF mice.** Quantification of aggregates size (**C**) and diameter (**D**) of the ASO (*n* = 3) and YF (*n* = 3) mouse at 8 months of age stained for α-synuclein using antibody against full-length α-synuclein (ASY6). Measurements performed in brain regions containing the globus pallidus (GP) and pyramis (PY) at the pontine level. Examples of aggregates are marked with arrowheads (**A**,**B**). When measured in terms of aggregate areas or diameters, aggregates are clearly larger in the YF mouse. Data are depicted as mean + standard deviation (SD); the variance did not differ between experimental groups and, thus, data was analyzed with one-way ANOVA with a multiple comparisons test. * *p* < 0.05, ** *p* < 0 .01. Scale bars: (**A**,**B**), 50 μm.

**Figure 8 cells-11-03673-f008:**
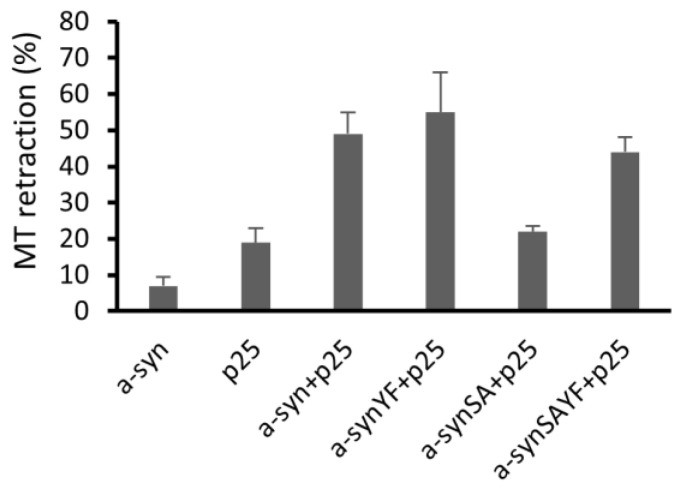
**Removal of tyrosine at positions 125, 133, and 136 of α-synuclein (YF) abrogates a protective effect of α-synuclein serine-129 to alanine (SA) mutation on microtubule (MT) retraction.** Cellular degeneration is quantified as the percentage of cells displaying MT retraction [23]. OLN-93 cells transiently transfected with α-synuclein or p25a, a-synSA + p25, a-synYF + p25, a-syn SAYF + p25 and subjected to immunofluorescence microscopy 24h post-transfection using antibodies against total α-synuclein and α-tubulin. By mutating the three tyrosines (YF), rescue of MT retraction otherwise obtained by the SA mutation is abrogated in this MT retraction assay. The data are representative of at least three independent experiments. Data are compared using one-way ANOVA followed by Sidak post hoc test for multiple comparisons.

## Data Availability

Not applicable.

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
