# Peer review of "Mutation of Tyrosine Sites in the Human Alpha-Synuclein Gene Induces Neurotoxicity in Transgenic Mice with Soluble Alpha-Synuclein Oligomer Formation"

_cells, 2022, doi:10.3390/cells11223673_

Round 1
Reviewer 1 Report
The authors present an interesting study related to mutations of tyrosine in the alpha-synuclein protein. This is important as posttranslational modifications are a key topic of interest concerning alpha-synuclein aggregation and toxicity. However, the severity of the phenotype in the transgenic mouse harbouring the tyrosine mutations (called YF mouse here) calls in to question the ethics of the study, specifically keeping the YF mouse alive and undergoing behavioural testing for several months. While I agree that it should be published so that no others attempt to do a similar thing with the same mouse, I would suggest major changes to the paper.
Minor comments
-Please make sure alpha-synuclein is in the same form throughout the manuscript
-Other abbreviations should be consistent throughout
-Proof-reading is needed, awkward sentences, grammar issues, particularly in the Methods and Discussion sections
-line 303 should be Snca
-Drosophila should be italicized and the same throughout
Comments
Introduction
-The first sentence should have references
-Paragraph starting at line 63 is confusing. You mention both phosphorylation of tyrosine and mutation to phenylalanine. I understand what you mean from checking the original article but it should be spelled out more clearly. Also, line 66: “and not at least in patients…” is awkward and should be rewritten to be more clear what you mean.
Methods
-Open field maze should be just open field
-Authors should clarify what is an open cage, is it an environment familiar to the mice? Then for YF mice the home cage is used
-Method for training on rotarod should be specified
-In 2.4 the method of sedation should be mentioned
Results
-Further to my above comment, were the mice compared on the open field in their home cage for all genotypes?
-The mice clearly exhibit a severe phenotype, which was previously also observed in Drosophila, therefore I question the long-term use of these animals (up to 7,5 months) when issues were already present earlier (1,5 months). It does not sound like the animals should have been subjected to later behavioural tests. I think that Figure 2 can almost be removed in its entirety and it only stated that their motor behaviour was so severely impaired.
-The biochemical and histological analysis is useful as it demonstrates what may be causing the phenotype, however I would prefer to see some quantification of neurons so that we could better understand why this is such a severe phenotype. This would add a lot to the study and also hopefully cover some ground so that others would not need to try to create these mice in order to quantify the potential neurodegeneration.
Discussion
-634-635: “…meaning that the cause of neurodegeneration was slowly going with blunt neuronal demise.” -This should be removed (or depending on above). There was no quantification of neuronal loss, and the YF mouse phenotype presentation was in no way “slow going”, it was quite fast.
-In relation to my main issue with the study (severe YF phenotype), the authors should add to the discussion how this does or does not relate to sporadic Parkinson’s disease, as it is difficult to see how this is clinically relevant, unless we are considering patients that would have a mutation in the tyrosine and present an early onset form of Parkinson’s
Author Response
Reviewer 1
Reviewer: The authors present an interesting study related to mutations of tyrosine in the alpha-synuclein protein. This is important as posttranslational modifications are a key topic of interest concerning alpha-synuclein aggregation and toxicity. However, the severity of the phenotype in the transgenic mouse harbouring the tyrosine mutations (called YF mouse here) calls in to question the ethics of the study, specifically keeping the YF mouse alive and undergoing behavioural testing for several months. While I agree that it should be published so that no others attempt to do a similar thing with the same mouse, I would suggest major changes to the paper.
> Our reply: We are pleased that the reviewer finds the manuscript of interest and importance. The concern on breeding and housing the YF mouse is appreciated, but as we state in the manuscript, the YF mice sometimes develop symptoms and pathology in a slower pace, and it was only in such situations that the mice were allowed to live for months. We have revised the sentence on this in the section 3.1.
Reviewer: Minor comments
-Please make sure alpha-synuclein is in the same form throughout the manuscript
-Other abbreviations should be consistent throughout
-Proof-reading is needed, awkward sentences, grammar issues, particularly in the Methods and Discussion sections
-line 303 should be Snca
-Drosophila should be italicized and the same throughout
> Our reply: We have gone through the manuscript and revised it accordingly including Snca in fig 1D.
Reviewer: Comments
Introduction
-The first sentence should have references
-Paragraph starting at line 63 is confusing. You mention both phosphorylation of tyrosine and mutation to phenylalanine. I understand what you mean from checking the original article but it should be spelled out more clearly. Also, line 66: “and not at least in patients…” is awkward and should be rewritten to be more clear what you mean.
> Our reply: We have revised these sentences.
Methods
-Open field maze should be just open field
-Authors should clarify what is an open cage, is it an environment familiar to the mice? Then for YF mice the home cage is used
-Method for training on rotarod should be specified
-In 2.4 the method of sedation should be mentioned
> Our reply: We have revised these sentences.
Results
-Further to my above comment, were the mice compared on the open field in their home cage for all genotypes?
-The mice clearly exhibit a severe phenotype, which was previously also observed in Drosophila, therefore I question the long-term use of these animals (up to 7,5 months) when issues were already present earlier (1,5 months). It does not sound like the animals should have been subjected to later behavioural tests. I think that Figure 2 can almost be removed in its entirety and it only stated that their motor behaviour was so severely impaired.
-The biochemical and histological analysis is useful as it demonstrates what may be causing the phenotype, however I would prefer to see some quantification of neurons so that we could better understand why this is such a severe phenotype. This would add a lot to the study and also hopefully cover some ground so that others would not need to try to create these mice in order to quantify the potential neurodegeneration.
> Our reply: We have revised these sentences. As mentioned above, not all mice were affected from an early age. We believe the Fig.2 is of importance for our data presentation and we politely refrain from removing it. We agree on the comment to the relevance of measuring neuronal loss in the YF mouse; we were not able to make this analysis as we were devoid of the sufficient number of mice needed to make a stereological measure.
Discussion
-634-635: “…meaning that the cause of neurodegeneration was slowly going with blunt neuronal demise.” -This should be removed (or depending on above). There was no quantification of neuronal loss, and the YF mouse phenotype presentation was in no way “slow going”, it was quite fast.
-In relation to my main issue with the study (severe YF phenotype), the authors should add to the discussion how this does or does not relate to sporadic Parkinson’s disease, as it is difficult to see how this is clinically relevant, unless we are considering patients that would have a mutation in the tyrosine and present an early onset form of Parkinson’s
> Our reply: We have revised and inserted new sentences.
Reviewer 2 Report
- Line 618, pg 20, is inconsistent with the data depicted in figure 1 D. No lifespan comparison between the YF and ASO mice was shown in the results. The authors must mention the lifespan difference between these two models in the results section.
- Figure 5: The distribution of full-length alpha-synuclein for the ASO mice was not included. Is there a reason for this?
- It is unclear why figure 6 B was included as it is not consistent with figure 3 findings demonstrating that YF young mice have lower phosphorylation in the synaptosome compared to the age-matched ASO model. Although figure 3B indicates that phosphorylated alpha-synuclein is higher in the cytosol of the YF model, the authors did not include the age for this specific experiment. It will be more consistent if the authors replace figure 6A with a figure representing ASO brains at 1.5 months as they did for the eight-month representative images. The WT 8 months results could be included in supplementary figures.
- Figure 7: A zoomed image of the examples of aggregates marked should be included.
Author Response
Reviewer 2
Reviewer: Comments and Suggestions for Authors
Line 618, pg 20, is inconsistent with the data depicted in figure 1 D. No lifespan comparison between the YF and ASO mice was shown in the results. The authors must mention the lifespan difference between these two models in the results section.
Figure 5: The distribution of full-length alpha-synuclein for the ASO mice was not included. Is there a reason for this?
It is unclear why figure 6 B was included as it is not consistent with figure 3 findings demonstrating that YF young mice have lower phosphorylation in the synaptosome compared to the age-matched ASO model. Although figure 3B indicates that phosphorylated alpha-synuclein is higher in the cytosol of the YF model, the authors did not include the age for this specific experiment. It will be more consistent if the authors replace figure 6A with a figure representing ASO brains at 1.5 months as they did for the eight-month representative images. The WT 8 months results could be included in supplementary figures.
Figure 7: A zoomed image of the examples of aggregates marked should be included.
> Our reply: We have revised these sentences. Concerning question 2 on ASO mice, we decided not to include these as they are intermediates between WT and YF mice. In fig 6B, we show YF rather than ASO to demonstrate the quite blunt appearance of phosphorylation at this early age, even in the YF with most pathology. Figure 3A+B represent an illustration of mice of mixed age and reveal that the a-synuclein levels are consistent over age. As stated above, the YF mice develop symptoms in different pace. In contrast, the a-synuclein levels in the synaptomes are highly dependent on age. Therefore, we think the 8 months observations are sufficiently relevant to be included among the regular illustrations. We do not think a zoom image should be included, as this is an option on the computer, which clearly elucidates the aggregates especially in the YF4 mouse.
Reviewer 3 Report
Dear Author,
Thanks for submitting your research manuscript entitled "Mutation of tyrosine sites in the human alpha-synuclein gene in transgenic mice induces neurotoxicity in vivo with soluble alpha-synuclein oligomer formation".
Before giving my final comments as well as the final revision of this manuscript, the author needs to address the following comments scientifically.
Major concerns:
Please find out the following comments:-
· The rationale and purpose behind selecting the selection of alpha-synuclein gene in association with neurotoxicity responses is explained very poorly, irrelevant and in incomplete manner throughout the manuscript.
· Lack of update references with incomplete experimental design is another major concern.
· Title, and abstract is misleading the reader. Title needs to reframe in simply manner accordingly.
· Rationale, Selection and evaluation of several cellular and molecular targets in association with memory defect is very poorly explained, and justified in abstract, intro as well in discussion part.
· The reviewer found irrational and non-scientific justification in the abstract—introduction and discussion part.
· Abstract is very poorly written and very confusing. Irrational and fused with repetitions. The reviewer found irrational and non-scientific justification in the abstract—introduction and discussion part.
Example 1: -synuclein with tyrosine mutated to phenylalanine at position 125 a: Overexpression of leads to a severe phenotype with motor impairment and neuropathology in Drosophila. Here, we hypothesized that tyrosine mutations would similarly lead to impaired motor performance with neuropathology in a mammalian model..????????????
Example 2: The YF mice had a decreased lifespan and displayed a dramatic motor –synuclein aphenotype with paralysis of both hind- and forelegs. Posttranslational modification of a due to phosphorylation of serine 129 is often seen in inclusions in the brains of patients with synucleinopathies?????????
Example 3: The study shows that the substitution of tyrosines to phenylalanine in α-synuclein at positions 125, 133, and 136 leads to severe toxicity in vivo. An indifferent change in phosphorylation of serine 129 upon tyrosine substitution suggests that the toxicity caused by the tyrosine mutations is not caused by serine 129 phosphorylation.???? What authors want to say? The incomplete justification and scientific correlation is another concern.
· The results and discussion are very poorly explained.
· Reviewer surprise to see the justification at the end of discussion part “This suggests that the tyrosines, potentially due to their phosphorylation, are not toxic on their own but they exhibit a permissive effect for the toxicity mediated by phosphorylated serine 129. The current study did not explore the possible neuroprotective potential of modulating serine and tyrosine phosphorylation in the C-terminus of α-synuclein, but collectively our in vivo and ex vivo data support a potential role in modulating α-synuclein dependent neurotoxicity.”Author need to directly strike in scientific and readily manner. And simplify whole manuscript directly focus on incidence of actual concern and remove all lines, paragraphs that are saying irrelevant correction etc……..
· The reviewer feels the author needs to elaborate and justify it with proper citations and strong evidence. The author fails to explain the relevant justification in the introduction as mentioned in the discussion part.
· A major drawback is a lack of supporting pre-clinical and clinical evidence regarding targeting drugs.
· Throughout the manuscript, the main focus is not clear
· Complete mismatch of abstract, introduction, results and discussion in concern with effective neuroprotective agents. Author didn’t justify specific.
Title:
· Mismatch of title with relevant introduction and conclusive remarks in the conclusion part.
Abstract:
- The rationale behind this research is not well explained, and several major concerns still constrain the reviewer's enthusiasm for publishing this manuscript.
Introduction:
- The basic literature is not well written and does not even include any literature on alternative approaches with updated references regarding involvement of current drug treatment/techniques used in association with alpha-synuclein related therapies or preventive measurements.
- Authors fail to justify the correlation, and almost irrational and common information is present in the introduction part.
Material and methods:
- Major drawback is the lack of supporting references and incomplete experimental and behavioral paradigms.
- All behavioral and few biochemical parameters are very poorly explained without any references.
- Provide all biochemicals kits numbers along with their city, country in all individual parameters in all expressions, blots, etc.
- In order to support the assessment of all mentioned parameters in his study, the author should provide all the source documents and data he/she has followed for all assays and estimates.
- How was the dosing determined? Dose-responses should be performed.
- How was the sample size determined? Ideally, a priori sample size calculation should be performed to determine the appropriate sample size.
- Normality and variance homogeneity should be assessed across all groups of the same outcome variable and not individual experimental groups. If the data were not normally distributed or variance homogeneity was not met, nonparametric tests need to be performed. Parametric data should be reported as mean +/- SD, while nonparametric data should be given/displayed as median and interquartile range. Longitudinal data should be analyzed using repeated measures tests.
Results:
- All results are very poorly explained. Revised all.
- All blot analysis, and (Figure 1b, Figure 3a,b,c,d) are highly blurred and there is no clarity for easy understanding. Not acceptable in current form.
- Re-check figures 4c, 3d, 2c and confirm either statistical symbol are properly mentioned in graphs or not?
- Results need more clarification and significant justification. Differentiating between the outcome and the discussion sections is quite difficult.
- High note: Must provide all results description and Use proper statistical reporting: i.e. for the results of each statistical test, the authors should report the statistical test that was applied, the test statistic (e.g. t, U, F, r), degrees of freedom as subscripts to the test statistic, and the exact probability value, including those for normality and variance homogeneity tests. Statistics should be reported in APA format, i.e.: t(df) = value, p = value; F(df1,df2) = value, p = value; r(df) = value, p = value; [chi]2 (df, N = value) = value, p = value; Z = value, p = value. Include statements on the tests for normality and variance heterogeneity and respective results. If the data were not normally distributed or variance heterogeneity was not met, nonparametric tests need to be applied.
Discussion:
- To address the outcome of in-vivo measures/results separately and how they correlate with the existing literature, it would be better if the author restructured to take a more critical approach for effective neuronal injury.
- In the discussion and the conclusion, the aims, rationale, and future perspectives are not evident clearly in relation with in-vitro and in-vivo experimentation.
- The discussion is usually unorganized at the beginning to address all the observations and evaluate them at the end. It makes the results easier to contextualize and simpler to comprehend.
- Furthermore, a minimal critical analysis should be provided, along with current study limitations as well the future perspective as separate paragraph.
Conclusion:
- Need to revise the conclusion in a scientific manner. Not accepted in its current form.
- This reviewer considers that this paper cannot be published in the present form. A detailed revision shortening, ordering and following the commented ideas could improve this interesting paper in a significant manner.
- Several typewriting mistakes are present and needing correction. This reviewer remains at entire disposal for the next version.
Author Response
Reviewer 3
Reviewer: Comments and Suggestions for Authors
Dear Author, Thanks for submitting your research manuscript entitled "Mutation of tyrosine sites in the human alpha-synuclein gene in transgenic mice induces neurotoxicity in vivo with soluble alpha-synuclein oligomer formation".
Before giving my final comments as well as the final revision of this manuscript, the author needs to address the following comments scientifically.
> Our reply: We find this extensive review of our manuscript very inappropriate, as it is loaded with incomplete sentences that often is difficult to read and are without clear suggestions for revision. Below follow our comments to the Reviewers marks.
Reviewer: Major concerns:
Please find out the following comments:-
The rationale and purpose behind selecting the selection of alpha-synuclein gene in association with neurotoxicity responses is explained very poorly, irrelevant and in incomplete manner throughout the manuscript.
Lack of update references with incomplete experimental design is another major concern.
> Our reply: We do not agree to these non-constructive, very general statements that preclude our possibilities for further revision.
Reviewer: Title, and abstract is misleading the reader. Title needs to reframe in simply manner accordingly.
Rationale, Selection and evaluation of several cellular and molecular targets in association with memory defect is very poorly explained, and justified in abstract, intro as well in discussion part.
The reviewer found irrational and non-scientific justification in the abstract—introduction and discussion part
Abstract is very poorly written and very confusing. Irrational and fused with repetitions. The reviewer found irrational and non-scientific justification in the abstract—introduction and discussion part.
Example 1: -synuclein with tyrosine mutated to phenylalanine at position 125 a: Overexpression of leads to a severe phenotype with motor impairment and neuropathology in Drosophila. Here, we hypothesized that tyrosine mutations would similarly lead to impaired motor performance with neuropathology in a mammalian model..????????????
Example 2: The YF mice had a decreased lifespan and displayed a dramatic motor –synuclein aphenotype with paralysis of both hind- and forelegs. Posttranslational modification of a due to phosphorylation of serine 129 is often seen in inclusions in the brains of patients with synucleinopathies?????????
Example 3: The study shows that the substitution of tyrosines to phenylalanine in α-synuclein at positions 125, 133, and 136 leads to severe toxicity in vivo. An indifferent change in phosphorylation of serine 129 upon tyrosine substitution suggests that the toxicity caused by the tyrosine mutations is not caused by serine 129 phosphorylation.???? What authors want to say? The incomplete justification and scientific correlation is another concern.
> Our reply: We have revised the abstract where we find it appropriate. We do not find the title misleading; but we have revised the title a bit for clarification. The “Example 1” mentioned by the reviewer is not understandable and we find the abstract text clearly understandable. It is not understandable what the reviewer suggests by “Example 2”. The “Example 3”: we have revised this sentence.
Reviewer: The results and discussion are very poorly explained.
- Reviewer surprise to see the justification at the end of discussion part “This suggests that the tyrosines, potentially due to their phosphorylation, are not toxic on their own but they exhibit a permissive effect for the toxicity mediated by phosphorylated serine 129. The current study did not explore the possible neuroprotective potential of modulating serine and tyrosine phosphorylation in the C-terminus of α-synuclein, but collectively our in vivo and ex vivo data support a potential role in modulating α-synuclein dependent neurotoxicity.”Author need to directly strike in scientific and readily manner. And simplify whole manuscript directly focus on incidence of actual concern and remove all lines, paragraphs that are saying irrelevant correction etc……..
- The reviewer feels the author needs to elaborate and justify it with proper citations and strong evidence. The author fails to explain the relevant justification in the introduction as mentioned in the discussion part.
- A major drawback is a lack of supporting pre-clinical and clinical evidence regarding targeting drugs.
- Throughout the manuscript, the main focus is not clear
- Complete mismatch of abstract, introduction, results and discussion in concern with effective neuroprotective agents. Author didn’t justify specific
> Our reply: We have revised the results and discussion sections where we find it appropriate. We have no problem with the sentences written in the last paragraph of the Discussion. Our manuscript does not deal with drugs or any sort of treatment approaches. We do not agree that the blottings shown in the illustrations are blurred.
> Our reply to the below-mentioned sentences:
We find these sentences highly inappropriate and refrain from further revision of these. Some of the sentences appear to be repeats of the sentences mentioned above.
(((((((((((((((((((((((((((((((((((((((((((((((((((((((((((((((((((((((((((((((((((((((((((((((((((((((((((((((((((((((((((((((((((((((((((((((((((((((((((
Title:
- Mismatch of title with relevant introduction and conclusive remarks in the conclusion part.
Abstract:
- The rationale behind this research is not well explained, and several major concerns still constrain the reviewer's enthusiasm for publishing this manuscript.
Introduction:
- The basic literature is not well written and does not even include any literature on alternative approaches with updated references regarding involvement of current drug treatment/techniques used in association with alpha-synuclein related therapies or preventive measurements.
- Authors fail to justify the correlation, and almost irrational and common information is present in the introduction part.
Material and methods:
- Major drawback is the lack of supporting references and incomplete experimental and behavioral paradigms.
- All behavioral and few biochemical parameters are very poorly explained without any references.
- Provide all biochemicals kits numbers along with their city, country in all individual parameters in all expressions, blots, etc.
- In order to support the assessment of all mentioned parameters in his study, the author should provide all the source documents and data he/she has followed for all assays and estimates.
- How was the dosing determined? Dose-responses should be performed.
- How was the sample size determined? Ideally, a priori sample size calculation should be performed to determine the appropriate sample size.
- Normality and variance homogeneity should be assessed across all groups of the same outcome variable and not individual experimental groups. If the data were not normally distributed or variance homogeneity was not met, nonparametric tests need to be performed. Parametric data should be reported as mean +/- SD, while nonparametric data should be given/displayed as median and interquartile range. Longitudinal data should be analyzed using repeated measures tests.
Results:
- All results are very poorly explained. Revised all.
- All blot analysis, and (Figure 1b, Figure 3a,b,c,d) are highly blurred and there is no clarity for easy understanding. Not acceptable in current form.
- Re-check figures 4c, 3d, 2c and confirm either statistical symbol are properly mentioned in graphs or not?
- Results need more clarification and significant justification. Differentiating between the outcome and the discussion sections is quite difficult.
- High note: Must provide all results description and Use proper statistical reporting: i.e. for the results of each statistical test, the authors should report the statistical test that was applied, the test statistic (e.g. t, U, F, r), degrees of freedom as subscripts to the test statistic, and the exact probability value, including those for normality and variance homogeneity tests. Statistics should be reported in APA format, i.e.: t(df) = value, p = value; F(df1,df2) = value, p = value; r(df) = value, p = value; [chi]2 (df, N = value) = value, p = value; Z = value, p = value. Include statements on the tests for normality and variance heterogeneity and respective results. If the data were not normally distributed or variance heterogeneity was not met, nonparametric tests need to be applied.
Discussion:
- To address the outcome of in-vivo measures/results separately and how they correlate with the existing literature, it would be better if the author restructured to take a more critical approach for effective neuronal injury.
- In the discussion and the conclusion, the aims, rationale, and future perspectives are not evident clearly in relation with in-vitro and in-vivo experimentation.
- The discussion is usually unorganized at the beginning to address all the observations and evaluate them at the end. It makes the results easier to contextualize and simpler to comprehend.
- Furthermore, a minimal critical analysis should be provided, along with current study limitations as well the future perspective as separate paragraph.
Conclusion:
- Need to revise the conclusion in a scientific manner. Not accepted in its current form.
- This reviewer considers that this paper cannot be published in the present form. A detailed revision shortening, ordering and following the commented ideas could improve this interesting paper in a significant manner.
- Several typewriting mistakes are present and needing correction. This reviewer remains at entire disposal for the next version)))))))))))))))))))))))))))))))))))))))))))))))))))))))))))))))))))))))))))))))))))))))))))))))))))))))))))))))))))))))))))))))))))))))))))))))))))
Round 2
Reviewer 1 Report
Some minor edits could still be done to the manuscript to make it more readable, but the authors have (somewhat) addressed my comments. I would still prefer to see some other neuronal quantification but understand it is not possible.
Author Response
Reviewer I asks for more information on quantitative changes in neuronal numbers in YF4 mice. We agree that this would have been interesting to count neurons but unfortunately, we were unable to perform a valid quantitative estimate based on the number of availble YF4 mice.
Reviewer 3 Report
Dear Author,
After careful revision, the revised manuscript can be proceed further for publication.
Author Response
We are pleased that reviewer 3 refrains from more comment to the manuscript.